# Are Semiquantitative Methods Superior to Deauville Scoring in the Monitoring Therapy Response for Pediatric Hodgkin Lymphoma?

**DOI:** 10.3390/jpm13030445

**Published:** 2023-02-28

**Authors:** Firuz Ibrahim, Michela Gabelloni, Lorenzo Faggioni, Subramanyam Padma, Arun R. Visakh, Dania Cioni, Emanuele Neri

**Affiliations:** 1Department of Nuclear Medicine and PET-CT, Burjeel Medical City, Abu Dhabi 92510, United Arab Emirates; 2Academic Radiology, Department of Translational Research, University of Pisa, 56126 Pisa, Italy; 3Amrita Institute of Medical Sciences, Kochi 682041, Kerala, India; 4Department of Nuclear Medicine, VPS Lakeshore Hospital, Kochi 682040, Kerala, India

**Keywords:** Hodgkin lymphoma, chemotherapy, positron emission tomography computed tomography (PET-CT), [18F]fluoro-2-deoxy-2-D-glucose (^18^F-FDG), Deauville score, standardized uptake value (SUV), response prediction, PET ratio (rPET)

## Abstract

Tailoring treatment in patients with Hodgkin lymphoma (HL) is paramount to maximize outcomes while avoiding unnecessary toxicity. We aimed to compare the performance of SUV_max_ reduction (ΔSUV_max%_) and the PET ratio (rPET) versus the Deauville score (DS) for assessing the chemotherapy response in pediatric HL patients undergoing ^18^F-FDG PET-CT. Fifty-two patients with biopsy-proven HL (aged 8–16 years) were enrolled at baseline, interim (after the second or third chemotherapy round) and post-therapy (on completion of first-line chemotherapy). Interim and post-therapy DS, ΔSUV_max%_ and rPET were compared as response predictors. Patients were classified as responders or non-responders based on a 24-month clinical follow-up. Interim DS showed a sensitivity, specificity, positive predictive value (PPV), negative predictive value (NPV) and diagnostic accuracy of 100%, 80.4%, 100%, 40% and 82.7%, respectively, in predicting the therapy response. Post-therapy DS showed a sensitivity, specificity, PPV, NPV and accuracy of 66.7%, 97.8%, 95.7%, 80% and 94.2%, repsectively. Interim ΔSUV_max%_ showed a sensitivity, specificity, PPV, NPV and accuracy of 83.3%, 82.6%, 97.4%, 38.5% and 82.7%, respectively, with a 56.3% cutoff. Post-therapy ΔSUV_max%_ showed a sensitivity, specificity, PPV, NPV and accuracy of 83.3%, 84.8%, 97.5%, 41.7% and 84.6%, respectively, with a 76.8% cutoff. Compared to ΔSUV_max%_, DS showed a significantly higher sensitivity, specificity (*p* < 0.05) and NPV (*p* < 0.01). The sensitivity, specificity, PPV, NPV and accuracy of rPET in predicting the therapy response at 24 months were 76.1%, 100%, 100%, 35.3% and 78.8%, respectively, with a cut-off of 1.31. DS and rPET showed comparable predictive performance (*p* > 0.58). In conclusion, DS is an easier method with better performance than ΔSUV_max%_ and rPET in predicting the chemotherapy response in pediatric HL patients.

## 1. Introduction

Hodgkin lymphoma (HL) is now one of the most curable forms of neoplasms in children. Compared to the past, survivors have a longer post-exposure life expectancy, but can experience long-term and latent side effects of cancer treatment in a non-negligible proportion of cases, especially in a post-radiotherapy setting, with the added risk of new malignancies in the long run [1,2,3,4,5,6]. Therefore, optimizing treatment to minimize subsequent risks while maintaining the chance of cure without compromising long-term survival is of utmost importance.

Fluorine-18 fluorodeoxyglucose positron emission tomography/computed tomography (^18^F-FDG PET-CT) is a highly sensitive tool for HL staging and restaging and is routinely used for response assessment and treatment adaptation. Various studies have demonstrated its role in preventing unnecessary side effects and morbidity and reducing treatment costs without compromising long-term survival [7,8,9].

^18^F-FDG uptake can be assessed using several approaches, including quantitative, semiquantitative and qualitative (i.e., visual) methods. While theoretically most accurate, the absolute quantification of ^18^F-FDG uptake is complex and impractical for routine clinical practice [10,11]. To mitigate this difficulty, semiquantitative approaches have been introduced and have become the standard of care in clinical practice [12]. The maximum standardized uptake value (SUV_max_) is a semiquantitative indicator of tumor ^18^F-FDG uptake and is the commonest method to assess tumor ^18^F-FDG concentration at a single point in time. However, SUV measurements are affected by multiple technical, physical and biological factors, e.g., the amount of radioactivity injected, tissue blood perfusion, blood sugar level, time of uptake, dose and PET-CT scanner calibration [13,14].

The Deauville score (DS) is a visual method based on a five-point scale that is commonly used in routine ^18^F-FDG PET-CT reporting and clinical trials for response assessment in HL and some non-HL tumors. It is based on the visual evaluation of PET images by comparing the ^18^F-FDG uptake in pathological sites to that of two reference points, such as the mediastinum and liver. However, DS relies on reader experience and knowledge of the physiologic FDG distribution and may be biased by optical distortion [15,16]. In addition, ^18^F-FDG uptake can also be detected in normal tissues (such as brown fat or rebound thymic hyperplasia), potentially leading to errors when the therapeutic response is evaluated using visual scoring methods [17].

SUV is routinely used in combination with visual assessment to assess the chemotherapy response in patients with HL. However, most extant studies on PET and the treatment response in patients with lymphoma have been performed in the adult population, and their findings have often been extended to pediatric patients without clear evidence whether the visual interpretation of ^18^F-FDG PET-CT images is adequate to reliably evaluate the treatment response. Since SUV calculation can be affected by the dose and body weight, many authors have suggested the utility of quantitative PET (qPET), determined as the ratio of the mean SUV of its four hottest voxels of the target lesion to the mean SUV of the liver [18]. Annunziata et al. proposed the PET ratio (rPET) as the ratio of SUV_max_ of the residual target lesion to the liver SUV_max_ [19]. However, these methods are still unused in routine clinical practice, and to our knowledge, little information exists on their use in the diagnostic workup of pediatric patients with HL, with no optimal cutoff value having been found so far.

Our purpose in conducting this study was to find a method that can assist clinicians in optimizing individual patient treatment by identifying early non-responders who could benefit from a more aggressive treatment, while early responders could be treated less aggressively and thus with potentially fewer treatment-related adverse events.

## 2. Materials and Methods

### 2.1. Patient Enrolment and ^18^F-FDG PET-CT Image Acquisition Protocol

This was a retrospective study involving 52 patients (male:female 26:26, age range 8–16 years, mean ± standard deviation 12 ± 2.4 years) with a histopathologic diagnosis of HL, who underwent ^18^F-FDG PET-CT at our referral center between January 2017 and January 2021. Written informed consent for PET-CT imaging was obtained from all patients, and institutional review board approval was waived due to the retrospective nature of our study. We examined ^18^F-FDG PET-CT examinations performed at baseline, after the second or third cycle of chemotherapy (interim) and after completion of first-line chemotherapy (post-therapy). All examinations were carried out on a commercial 16-row PET-CT whole-body scanner (GE Discovery 610^®^, General Electric, Milwaukee, WI, USA). Patients fasted for 6–8 h before intravenous administration of ^18^F-FDG (5.18–7.4 MBq/kg body weight) to reach a serum glucose level below 180 mg/dL. PET-CT images with a longitudinal coverage from the proximal thighs to the skull base were acquired 45–60 min after ^18^F-FDG injection. Images were reconstructed with a 128 × 128 matrix using an ordered subset expectation maximum iterative reconstruction algorithm, an 8 mm Gaussian filter and a field of view of 50 cm. CT, PET and co-registered PET-CT images were reviewed as source axial images, as sagittal and coronal reformations and using maximum-intensity projection views by two nuclear medicine physicians with 20 and 4 years of experience. Any disagreement between the two readers was resolved by reaching a consensus. Patient medical history, clinical examination and lab results (including LDH and bone marrow tests, if available) were accessed from our hospital information system. HL staging was performed using the Ann Arbor staging system.

The distribution of patients according to HL subtype and stage is reported in Table 1 and Table 2, respectively. All patients received chemo- and/or radiotherapy depending on the disease stage.

### 2.2. Visual and Quantitative Assessment of ^18^F-FDG PET-CT Images

Increased ^18^F-FDG uptake in nodes or at extranodal sites was considered as a positive ^18^F-FDG PET-CT finding. Care was taken to avoid areas of ^18^F-FDG uptake related to any physiological ^18^F-FDG activity. All positive ^18^F-FDG-avid lesions were annotated and followed up at interim and post-treatment PET-CT examinations. Interim and post-therapy PET-CT scans were evaluated visually using the Deauville five-point scoring system, and for each lesion, the maximum DS was recorded. Scores were computed based on current Lugano criteria, which categorize residual ^18^F-FDG uptake with respect to the mediastinum and liver (Table 3).

Complete response is defined as complete normalization of ^18^F-FDG uptake (DS from 1 to 3) [20]. Patients with increased ^18^F-FDG uptake of pathological lesions compared to the liver (DS higher than 3) were considered as poor responders based on visual scoring.

SUV_max_ was calculated on a volume of interest placed semiautomatically on each node or at each extranodal site with pathologically increased ^18^F-FDG uptake, by using the following formula:$${\mathrm{SUV}}_{\max}\;\left(g/\mathrm{mL}\right)=\frac{\mathrm{maximum}\;\mathrm{tumor}\;\mathrm{activity}\;\mathrm{concentration}\;\left(\mathrm{mCi}/\mathrm{mL}\right)\;•\;\mathrm{body}\;\mathrm{weight}\;\left(g\right)}{\mathrm{injected}\;\mathrm{dose}\;\left(\mathrm{mCi}\right)}$$

For each patient, the lesion with the highest SUV_max_ was recorded, and the follow-up value for the same lesion was used to calculate the percentage change in SUV_max_ (ΔSUV_max_) using the formula:$${\Delta \mathrm{SUV}}_{\max\%}=100\;•\;\frac{\mathrm{highest}\;\mathrm{SUV}\;\left({\mathrm{PET}}_{\mathrm{baseline}}\right)-\;\mathrm{highest}\;\mathrm{SUV}\;\left({\mathrm{PET}}_{\mathrm{interim}}\right)\;}{\mathrm{highest}\;\mathrm{SUV}\;\left({\mathrm{PET}}_{\mathrm{baseline}}\right)}$$

Finally, rPET was calculated as the ratio between the target lesion SUV_max_ and liver SUV_max_ at interim PET.

Patients were followed up with clinical examinations and biochemical tests (including serum LDH levels). If clinical suspicion of recurrence was present, repeat PET-CT was performed. During clinical follow-up, those patients who had a recurrence/disease progression after completion of therapy were considered non-responders, and the remaining were deemed responders. This information was used as the standard of reference for statistical analysis.

Patients were classified as true positive at follow-up if they had evidence of disease at interim imaging, followed by persistent disease and/or evidence of recurrence at the end of therapy. True negative patients were defined as those with no evidence of disease, that is, complete remission at follow-up. False-positive patients showed evidence of disease at interim PET-CT imaging but remained in remission during follow-up. Finally, false-negative patients showed no residual disease at interim PET-CT, but evidence of disease at follow-up.

### 2.3. Statistical Analysis

Data were analyzed using the SPSS software package, version 20.0 (https://www.ibm.com/products/spss-statistics). Categorical and quantitative variables were expressed as the frequency (percentage) and mean ± standard deviation, respectively. Receiver operating characteristic (ROC) curves were generated, and the area under the ROC curve was calculated to assess the diagnostic accuracy of ΔSUV_max%_ in detecting a positive response, along with the corresponding cut-off scores. Sensitivity, specificity, positive predictive value (PPV), negative predictive value (NPV) and diagnostic accuracy were calculated as indicators of the performance of DS and ΔSUV_max%_ in predicting positive response. Spearman’s rank test was used to find a correlation between DS and ΔSUV_max%_. Finally, the z-test was used to compare the performance of DS and ΔSUV_max%_ in predicting a positive response in all patients. A *p*-value less than 0.05 was set as the threshold for statistical significance.

Receiver operating characteristic (ROC) curves were generated, and the area under the ROC curve was calculated to assess the diagnostic accuracy of rPET in detecting a positive response, along with the corresponding cut-off scores. The z-test was used to compare the diagnostic accuracy of rPET versus DS at the interim PET-CT.

## 3. Results

### 3.1. Baseline SUV_max_ in Predicting a Positive Response

Of the 52 patients, 46 (88.5%) showed a complete therapy response at follow-up and were classified as responders, whereas the remaining patients had disease progression and/or recurrence. Baseline SUV_max_ was 12.3 ± 5.7 g/mL (mean ± standard deviation; range 3.2–29.8 g/mL), interim SUV_max_ was 3.8 ± 4.4 g/mL (range 1.2–21.6 g/mL) and post-therapy SUV_max_ was 2.1 ± 1.9 g/mL (range 1.2–9.0 g/mL).

When using a cut-off of 15.5 g/mL, baseline SUV_max_ for predicting a positive response to therapy at 24 months showed a sensitivity of 80.4%, specificity of 66.7%, PPV of 94.9%, NPV of 30.8% and accuracy of 78.8% (Figure 1).

When using the DS method, 31 (71.1%) patients were classified as responders (DS 1, 2 and 3) at interim PET-CT, whereas 47 (90.4%) were classified as responders at post-therapy PET-CT. No patients had a DS of 5 at either the interim or post-therapy PET-CT. Five (9.6%) patients had a DS of 4 after completion of therapy and were treated more aggressively with more cycles of chemotherapy or a change in the regimen and/or field radiotherapy.

When using the semiquantitative method, interim ΔSUV_max%_ was 69.1 ± 24.8 g/mL, whereas post-therapy ΔSUV_max%_ was 81.7 ± 11.4 g/mL.

### 3.2. Performance of ΔSUV_max%_ in Predicting a Therapy Response at 24 Months

For ΔSUV_max%_ at the interim or post-therapy PET-CT, an optimal cut-off of 56.3% or 76.8% was found, respectively (Figure 2).

The sensitivity, specificity PPV, NPV and accuracy of interim ΔSUV_max%_ in predicting the therapy response at 24 months were 83.3%, 82.6%, 97.4%, 38.5% and 82.7%, respectively.

The sensitivity, specificity PPV, NPV and accuracy of post-therapy ΔSUV_max%_ in predicting the therapy response at 24 months were 83.3%, 84.8%, 97.5%, 41.7% and 84.6%, respectively.

### 3.3. Diagnostic Accuracy of Prognostication by Deauville Criteria (Visual Assessment) at Different Intervals of Time in the Prediction of a Response at 24 Months

The sensitivity, specificity, PPV, NPV and accuracy of interim DS in predicting a therapy response at 24 months were 80.4%, 100.0%, 100.0%, 40.0% and 82.7%, respectively.

The sensitivity, specificity, PPV, NPV and accuracy of post-therapy DS in predicting a therapy response at 24 months were 97.8%, 66.7%, 95.7%, 80.0% and 94.2%, respectively.

Table 4 compares the diagnostic performance of ΔSUV_max%_ and DS at the interim PET-CT in predicting a therapy response at 24 months. Compared with interim ΔSUV_max%_, interim DS yielded a significantly higher specificity (100% vs. 83.3%, *p* = 0.002) and lower false-positive rate (0% vs. 16.7%, *p* = 0.002).

Compared with post-therapy ΔSUV_max%_, post-therapy DS yielded a higher sensitivity (97.8% vs. 84.8%, *p* = 0.018), specificity (83.3% vs. 67.7%, *p* = 0.049) and NPV (80.0% vs. 41.7%, *p* = 0.001), and lower false-positive (16.7% vs. 33.3%, *p* = 0.049) and false-negative rates (2.2% vs. 15.2%, *p* = 0.018) (Table 5).

Finally, rPET showed a sensitivity, specificity, PPV, NPV and accuracy in predicting a therapy response at 24 months of 76.1%, 100%, 100%, 35.3% and 78.8%, respectively, with a cut-off of 1.31 and an AUC of 0.913 (CI_95%_ 0.826–1, *p* = 0.0001). The predictive performance of rPET was comparable to that of DS (Table 6). 

## 4. Discussion

Hodgkin lymphoma is the third-most-common childhood malignancy. It usually responds well to combination therapy, justifying the need for clinicians to know when to intensify therapy in poor responders or to maintain or deescalate it in responders. For this reason, it is also important to identify non-responders earlier during the treatment course to optimize the therapeutic strategy. For that purpose, the role of ^18^F-FDG PET-CT in the management of adult HL patients is well-known and widely accepted [20,21,22,23].

We enrolled 52 patients, a relatively large number for a single-center study carried out in a high-volume referral center and in a context where HL incidence is lower in our region compared to other regions of our country (BLINDED) [24]. However, when compared to other studies conducted on the same subject, the sample size was generally low [10,24,25,26,27].

Accurate HL staging is the most important factor for setting a prognosis and deciding treatment options. The addition of ^18^F-FDG PET-CT can help identify disease locations that could be missed by CT alone [19]. In our study, most patients had stage 2A (28.8%) and stage 3A (21.2%) categories, with the commonest histopathologic type being the nodular sclerosing variant (63.4%).

### 4.1. Semiquantitative Assessment

Based on our findings, we recommend that pediatric patients with SUV_max_ higher than 15.5g/mL at baseline ^18^F-FDG PET-CT should undergo a more stringent follow-up. However, it is known that SUV measurements can be affected by several factors that may result in considerable variations of accuracy and reproducibility, including alterations in the calibration of the PET scanner or dose calibrator, tracer extravasation at the injection site, elevated blood glucose levels or patient motion (leading to SUV measurement errors up to 50%) and the partial volume effect (which may lead to SUV underestimation in smaller tumors). It has been found that both liver and mediastinal blood pool SUV could be predicted by patient weight [11,28,29]. Malladi et al. showed that liver SUV was affected by gender, whereas mediastinal SUV was dependent on the uptake time [21]. Additional studies with a larger sample size should be conducted to corroborate our findings.

Unlike for adults, to our knowledge, ΔSUV_max%_ cut-off values validated from large trials are currently unavailable for differentiating good from poor pediatric responders, with some data having only been provided by a few studies based on small patient cohorts [27,30]. Our findings showed that a 56.3% cut-off for interim ΔSUV_max%_ yielded a sensitivity, specificity and accuracy higher than 80% and a PPV higher than 95%. In a prospective study by Furth et al. on pediatric HL, a similar cut-off (58%) for ΔSUV_max%_ was used for predicting disease relapse using interim ^18^F-FDG PET-CT, yielding a sensitivity, specificity, NPV and accuracy of 100%, 97%, 100% and 97%, respectively [27]. Based on these findings, patients with ΔSUV_max%_ less than 56% at interim ^18^F-FDG PET-CT are more likely to be non-responders or at a higher risk of relapse at follow-up.

The usefulness of interim PET-CT in predicting a treatment response is underscored by its high PPV. Based on our findings, patients with an interim ΔSUV_max%_ lower than 56% should be followed up more aggressively. Of note, the fact that all parameters that can affect SUV calculation, and thus will also affect the ΔSUV_max%_ calculation, should be considered.

### 4.2. Deauville Scoring

Our findings showed that DS at the interim PET-CT had a specificity and PPV of 100% in predicting a treatment response at 24 months, higher than in previous studies [25,27,29]. One explanation could be that the older studies followed heterogeneous visual criteria, which might have affected the prognostic accuracy of the interim PET-CT, as pointed out by Terasawa et al. [31]. Our finding that DS at both the interim and post-therapy PET-CT can predict a response with a high PPV seems to suggest a more aggressive treatment for patients with a score higher than 3, as opposed to those with a lower score. This is in line with the results by Ilivitzki et al., who found that visual interpretation has a higher PPV in the early evaluation of chemosensitivity in pediatric HL patients [32].

Care must be taken when DS is performed as the ^18^F-FDG distribution in pediatric patients is slightly different compared to adults. It is known that pediatric patients may show a thymic rebound at the post-therapy PET-CT, which can be misleading. Moreover, liver tissue in pediatric patients is less affected by fatty changes than in adults, and it has been proven that ^18^F-FDG metabolism may be altered in patients with liver cirrhosis, potentially affecting both visual scoring and SUV measurements. Thermogenic brown adipose tissue, which is commonly seen in pediatric patients, can affect the quality of PET images, and knowledge of its normal distribution can help avoid false-positive findings [33]. Visual ^18^F-FDG uptake comparison can be optically distorted due to different background levels (simultaneous contrast illusion), as proposed by Meigan et al. [15,33,34,35]. In children aged between 6 and 8 years, intense ^18^F-FDG uptake can be seen within pharyngeal and palatine tonsils, potentially causing difficulties in interpretation [36]. Moreover, new diffuse ^18^F-FDG uptakes in the bone marrow and spleen can also be seen after chemotherapy. The aforementioned factors were considered when interpreting interim PET-CT scans.

### 4.3. rPET and DS

Based on our findings, patients can be classified as good or poor responders based on having an rPET value lower or higher than 1.31, respectively, with a sensitivity of 76.1% and a specificity of 100%. Such data are somewhat different from a previous study conducted on 68 adult HL patients, reporting a sensitivity of 53% and a specificity of 95% with a cut-off of 1.14 [19]. Another study performed on pediatric patients with extranodal lymphomas reported a cut-off of 1.25 with a sensitivity of 91.7% and specificity of 100% [37]. Further investigations are warranted on larger cohorts of pediatric patients to corroborate our findings.

In line with previous studies [19,37], DS and rPET showed a comparable performance in predicting a therapy response, characterized by a high specificity and PPV and a slightly higher sensitivity for DS. This suggests that the use of rPET did not yield any significant advantage over DS for the prediction of a therapeutic response.

### 4.4. Comparison of DS and SUV_max%_ Reduction

For interim PET-CT, a statistically significant association was found in the specificity and false-positive rates when ΔSUV_max%_ and DS were compared, and the latter showed a better performance (83.3% vs. 100% and 16.7% vs. 0%, respectively), whereas the remaining parameters were concordant. Our finding reflects the superiority of DS at the interim PET-CT. In contrast, a similar retrospective study by Ferrari et al. found DS to be inferior to ΔSUV_max%_, but it was based on a smaller sample size (N = 30) [25]. The main reason for this was the higher false-positive rate for DS due to the presence of inflammatory cells in the tumor microenvironment and the factors that affect background ^18^F-FDG uptake [24,28].

Moreover, a fixed ΔSUV_max%_ cannot always be used when the baseline SUV_max_ is low (e.g., <10 g/mL) or when SUV_max_ in the residual lesion is >5 g/mL. In such instances, both the DS and quantitative ΔSUV_max%_ methods are needed, as suggested at the Third International Workshop on Interim Positron Emission Tomography in Lymphoma [16].

### 4.5. Study Limitations

One limitation of our study was its small sample size and relatively short follow-up time. Another limitation related to the former is the small NPV obtained for ΔSUV_max%_ and rPET, likely due to the small number of nonresponders in the context of a low overall sample size. Furthermore, the retrospective nature of our study may have introduced some inconsistencies in the ^18^F-FDG doses injected and scan timing. Prospective studies on larger patient samples should provide more robust results.

## 5. Conclusions

Compared to the ΔSUV_max%_ method, Deauville scoring is an easier method yielding better specificity and a positive predictive value at interim PET-CT imaging for the assessment of the treatment response in pediatric patients with HL. rPET with a cut-off of 1.34 showed a comparable predictive performance to DS. Such findings seem to suggest that DS should be the preferred method for use in clinical practice. Moreover, patients with higher DS at the interim and post-therapy PET-CT should undergo a more stringent follow-up as they have a higher chance of treatment failure or disease relapse. Conversely, ΔSUV_max%_ should be used in patients classified as DS 4 or 5 at the interim PET-CT. In such instances, patients with a SUV_max%_ reduction less than 56% should be treated more aggressively. The issue of a low negative predictive value for ΔSUV_max%_ should be evaluated in a larger multicenter trial.

## Figures and Tables

**Figure 1 jpm-13-00445-f001:**
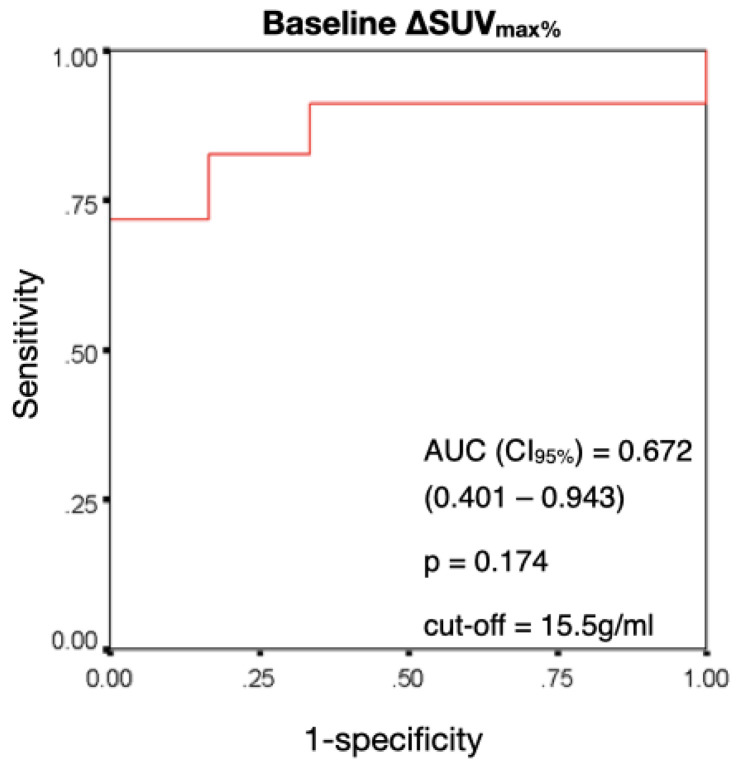
ROC curve for SUV_max_ prediction of response at 24 months at baseline ^18^F-FDG PET-CT. AUC = area under the ROC curve. CI_95%_ = 95% confidence interval.

**Figure 2 jpm-13-00445-f002:**
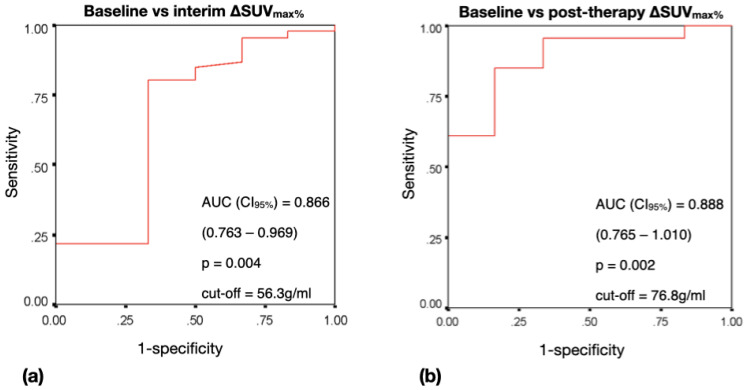
ROC curves for therapy response prediction at 24 months based on ΔSUV_max%_ at interim (**a**) and post-therapy PET-CT (**b**).

**Table 1 jpm-13-00445-t001:** Percentage distribution of the patient sample according to HL subtype.

HL Subtype	Patients (N)	Proportion (%)
Mixed cellularity	9	17.3
Nodular lymphocyte predominant	8	15.4
Nodular sclerosing	35	67.3

**Table 2 jpm-13-00445-t002:** Percentage distribution of the patient sample according to HL stage.

HL Stage	Patients (N)	Proportion (%)
1A	10	19.2
1B	3	5.8
2A	15	28.8
2B	5	9.6
3A	11	21.2
3B	3	5.8
4A	3	5.8
4B	2	3.8

**Table 3 jpm-13-00445-t003:** Modified Lugano five-point scale [20].

Score	Description
1	No uptake
2	Uptake ≤mediastinum
3	Uptake >mediastinum but ≤liver
4	Uptake moderately increased above liver at any site
5	Markedly increased uptake above the liver at any site
NE	Not evaluable
X	Any areas of uptake not likely to be related to lymphoma
1	No uptake

**Table 4 jpm-13-00445-t004:** Comparison of ΔSUV_max%_ and DS performance levels at interim PET-CT in predicting a therapy response at 24 months. Values are expressed as percentages.

	ΔSUV_max%_	DS	z (*p*)
Sensitivity	82.6	80.4	0.286 (0.779)
Specificity	83.3	100.0	3.075 (0.002)
False-negative rate	17.4	19.6	0.286 (0.779)
False-positive rate	16.7	0.0	3.075 (0.002)
PPV	97.4	100.0	1.162 (0.246)
NPV	38.5	40.0	0.161 (0.873)
Positive likelihood ratio	5.0	-	- (-)
Negative likelihood ratio	0.2	0.2	0.015 (0.992)
Accuracy	82.7	82.7	0 (1)

**Table 5 jpm-13-00445-t005:** Comparison of ΔSUV_max%_ and DS performance levels at post-therapy PET-CT in predicting a therapy response at 24 months. Values are expressed as percentages.

	ΔSUV_max%_	DS	z (*p*)
Sensitivity	84.8	97.8	2.36 (0.018)
Specificity	66.7	83.3	1.963 (0.049)
False-negative rate	15.2	2.2	2.36 (0.018)
False-positive rate	33.3	16.7	1.963 (0.049)
PPV	97.5	95.7	0.495 (0.624)
NPV	41.7	80.0	4.004 (0.001)
Positive likelihood ratio	5.1	2.9	0.559 (0.582)
Negative likelihood ratio	0.2	0.0	0.233 (0.818)
Accuracy	84.6	94.2	1.594 (0.112)

**Table 6 jpm-13-00445-t006:** Comparison of rPET and DS performance levels at interim PET-CT in predicting a therapy response at 24 months. Values are expressed as percentages.

	rPET	DS	z (*p*)
Sensitivity	76.1	80.4	0.537 (0.596)
Specificity	100.0	100.0	-
False-negative rate	23.9	19.6	0.548 (0.589)
False-positive rate	0.0	0.0	-
PPV	100.0	100.0	-
NPV	35.3	40.0	0.519 (0.610)
Positive likelihood ratio	-	-	-
Negative likelihood ratio	0.2	0.2	0.051 (0.960)
Accuracy	78.8	82.7	0.535 (0.596)

## Data Availability

The data can be made publicly available on reasonable request.

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
