# Peer review of "Are Semiquantitative Methods Superior to Deauville Scoring in the Monitoring Therapy Response for Pediatric Hodgkin Lymphoma?"

_jpm, 2023, doi:10.3390/jpm13030445_

Round 1

Reviewer 1 Report

Ibrahim et al. in their retrospective study evaluate the value of semiquantitative evaluation (using ΔSUVmax%) compared to visual analysis (using Deauville score) of [18F]FDG PET/CT in the treatment response evaluation of pediatric HL.

The authors pointed out the parameters that could impact treatment patients' management underlining the need for adapted therapy in patients with positive interim or end-of-treatment PET/CT scans. Considering the limited literature about the topic in the pediatric setting, I think there is value to publish but I do think additional revisions and/or questions to address remain.

Abstract:

-        Please spell out all acronyms in the abstract

Introduction:

-        Concerning the pediatric sample, I think that some more extensive observations about the role of Deauville and semiquantitative analysis in this setting of patients deserve more background.

-        The purpose of the study is not well described, and I think the sentence “Our purpose is to develop a method that can assist deciding whether therapy regimens could be tapered to reduce the risk of toxicity” is general and some details could be added.

Material and methods:

-        Please describe clinical parameters collected for each patient

-        Please correct “8F-FDG” in line 103.

-   Lugano criteria should be described in detail as well as Deauville 5-point scale.

-        The sentence “Unlike for adults, to our knowledge ΔSUVmax cut-off values validated from large trials for differentiating good from poor pediatric responders are currently unavailable, with some data having only been provided by a few studies based on small patient cohorts [19-20]” (line 121-123) is more appropriate in the discussion session.

-        How was the clinical follow-up assessment performed?

-        How was the prediction of response using the DS and ΔSUVmax correlated with disease-free survival (line 120)?

Results:

-        If available, it would be interesting to add chemotherapy and/or radiotherapy regimen for each patient.

-        Please check the abbreviations below the table and figure.

-        According to the endpoint of the study, it is suitable to describe in detail the chemotherapy regimen and/or field radiotherapy (line 164-165) used in the DS 4 patients after the evaluation.

Discussion:

-        I think that is important to discuss the low NPV results among the used scales.

-        Please introduce study limitations.

Overall, the English and grammar are good, there are some issues regarding sentence formation, word usage, and grammar, which could benefit from further review.

Reviewer 2 Report

The manuscript describes a study comparing SUVmax and changes in SUVmax during therapy as predictor of response to therapy in pediatric hodgkin lymphoma and compares this to Deauville score.

There are some general issues to address. First of all, as the authors state themselves, absolute SUVs are affected by a number of variables (blood glucose, uptake time etc), so why do they choose this as their predictor? Why not use a reference tissue as liver or mediastinal blood pool to normalize the SUVs?

Which brings me to the next point. qPET as a quantitative extension of Deauville score has been described in large multicenter studies of pediatric HL. Why do the authors not include this in their analysis? Or even mention it?

Another issue to address is the sample size. Although the total number of patients is 52, which is not a lot, but could be considered acceptable in some analyses, the number of patients with progressive/recurrent disease was only 6. This is simply not enough for making robust ROC analyses.

I have a few specific comments as well:

Introduction:

Nomenclature of [18F]FDG - please adhere to guidelines, for instance: https://www.eanm.org/publications/guidelines/nomenclature/

P 1 line 65: 18F-FDG uptake in brown fat is seen in adults as well.

Results:

How do you select the cut off values for SUVmax and ΔSUVmax?

Could you please provide 95% confidence intervals for sensitivity, specificity etc)

P. 5 line 181-182 + Table 3. There is a discrepancy here. In the text the sensitivity and specificity of interim DS is 100% and 80.4%, respectively. In the table, the sensitivity is 80.4% and specificity is 100%. Which is correct?

The same mistanke is seen for post-therapy DS I line 183-184 and Table 4.

Reviewer 3 Report

Dear authors, a good job, well designed.

However, I think that the purpose of the research should be clearly specified in the introduction.

I also think that the most recent work on the subject should be mentioned in the discussion. Specifically, I found a very recent research by an Egyptian group, here enclosed, which compares the Deauville score with the ratio between target lesion and liver maximum standardized uptake values. In my opinion these results should be included and commented on, in comparison with yours.

Best regards

Prof. Lucio Mango

Round 2

Reviewer 1 Report

the authors modified the manuscript as suggested.